# Research Evidence on High-Fat Diet-Induced Prostate Cancer Development and Progression

**DOI:** 10.3390/jcm8050597

**Published:** 2019-04-30

**Authors:** Shintaro Narita, Taketoshi Nara, Hiromi Sato, Atsushi Koizumi, Mingguo Huang, Takamitsu Inoue, Tomonori Habuchi

**Affiliations:** Department of Urology, Akita University School of Medicine, Akita 010-8543, Japan; bkspt512@yahoo.co.jp (T.N.); hiromisato2002@yahoo.co.jp (H.S.); koizu3atsu4@yahoo.co.jp (A.K.); huangmg0319@yahoo.co.jp (M.H.); takmitz@gmail.com (T.I.); thabuchi@gmail.com (T.H.)

**Keywords:** animal model, diet, fat, in vitro, in vivo, mouse, prostate cancer

## Abstract

Although recent evidence has suggested that a high-fat diet (HFD) plays an important role in prostate carcinogenesis, the underlying mechanisms have largely remained unknown. This review thus summarizes previous preclinical studies that have used prostate cancer cells and animal models to assess the impact of dietary fat on prostate cancer development and progression. Large variations in the previous studies were found during the selection of preclinical models and types of dietary intervention. Subcutaneous human prostate cancer cell xenografts, such as LNCaP, LAPC-4, and PC-3 and genetic engineered mouse models, such as TRAMP and Pten knockout, were frequently used. The dietary interventions had not been standardized, and distinct variations in the phenotype were observed in different studies using distinct HFD components. The use of different dietary components in the research models is reported to influence the effect of diet-induced metabolic disorders. The proposed underlying mechanisms for HFD-induced prostate cancer were divided into (1) growth factor signaling, (2) lipid metabolism, (3) inflammation, (4) hormonal modulation, and others. A number of preclinical studies proposed that dietary fat and/or obesity enhanced prostate cancer development and progression. However, the relationship still remains controversial, and care should be taken when interpreting the results in a human context. Future studies using more sophisticated preclinical models are imperative in order to explore deeper understanding regarding the impact of dietary fat on the development and progression of prostate cancer.

## 1. Introduction

Prostate cancer is the most common type of cancer among men in 92 countries and the leading cause of cancer deaths among men in 48 countries [1]. In the United States as well, prostate cancer has been the most commonly diagnosed type of cancer among men, accounting for almost 1 in 5 new diagnoses [2]. While the incidence of latent prostate cancer has been similar between the United States and Japan, the incidence of clinically detected prostate cancer has been lower in Asia, including Japan [3,4]. Of note, the incidence of prostate cancer in Chinese and Japanese men has been reported to increase substantially after migration to the United States [5]. Furthermore, the morbidity and mortality due to prostate cancer in Asia increased remarkably in recent years [6]. Although the etiology of prostate cancer is multifocal, these epidemiological findings, including geographic and ethnic differences, suggest that lifestyle and/or environmental factors have a substantial influence on the development and progression of prostate cancer [7]. Epidemiological evidence suggested that among the acquired risk factors for prostate cancer development and progression, diet and obesity have a potential to cause prostate cancer initiation, promotion, and progression [8,9]. Several studies have implicated dietary fats as important factors of prostate cancer risk and its aggressive phenotype [9,10]. A number of clinical and preclinical studies have shown that total fat intake and specific fat composition play a potential role in prostate cancer, although their findings have remained inconclusive.

Considering these backgrounds, this study aimed to summarize previous preclinical studies regarding the relationship between dietary fat and prostate cancer development and progression, focusing on differences in preclinical models and dietary fat composition. Furthermore, potential mechanisms on dietary fat-induced prostate carcinogenesis were discussed by updating previous research evidence. To this end, previous preclinical studies investigating dietary fat and prostate cancer were identified using a PubMed search including only studies published in English. This review helps us to understand the current state of diet-induced prostate cancer research in order to guide future works exploring the association between dietary-fat and prostate cancer.

## 2. Various Preclinical Models

A number of animal models, including those involving prostate cancer cell xenografts and allografts, Transgenic Adenocarcinoma of the Mouse Prostate (TRAMP) mice, and other genetically engineered mice targeting oncogenes and tumor suppressor genes, were tested in order to assess the impact of dietary-fat intake on prostate cancer development and progression (Table 1). First, the models used in the previous studies were summarized as follows.

### 2.1. Human Cancer Cell Xenograft and Allograft Models

The most experienced models to assess the impact of dietary fats on prostate cancer growth were subcutaneous xenograft models [11,12,15,16,35,52,58].

Nude [16] and severe combined immunodeficient (SCID) mice [15] were frequently used as host mice for human prostate cancer cell xenografts. In 1995, Wand et al. first assessed the impact of five different fat percentages on human prostatic adenocarcinoma (LNCaP) xenograft growth using athymic nude mice [11]. Accordingly, mice who continued to receive a 40.5-kcal% fat diet had substantially greater tumor growth rates, final tumor weights, and final tumor weight to animal weight ratios compared to those whose diets were changed to 2.3 kcal%, 11.6 kcal%, or 21.2 kcal% fat, suggesting that those fed low-fat diets (LFDs) had decreased growth of established LNCaP tumors. An additional study demonstrated that an isocaloric LFD (12 kcal% fat) resulted in significantly slower tumor growth rates and lower serum prostate-specific antigen (PSA) levels compared to a high-fat diet (HFD) using LAPC-4 xenografts on SCID mice [15]. The same group also showed that reduced dietary fat intake delayed conversion from androgen-sensitive to androgen-insensitive prostate cancer and significantly prolonged survival of SCID mice bearing LAPC-4 xenografts [58]. Moreover, we had previously found that Balb/c-nu/nu mice receiving a HFD had significantly higher LNCaP xenograft tumor volumes and serum PSA levels than those receiving an LFD [52]. The impact of a HFD on xenograft tumor growth using other human prostate cancer cell lines, such as 22Rv-1 and PC-3, had also been investigated in previous literatures [32,51]. Although the significance of the effect varied, a number of studies proposed that a HFD accelerated tumor growth of human prostate cancer cell xenografts inoculated into immunodeficient mice. Conversely, several studies have found no relationship between a HFD and xenograft growth [22,26]. In a study comparing LAPC-4-xenografted SCID mice receiving an isocaloric Western diet (40% fat and 44% carbohydrate) and those receiving an LFD (12% fat and 72% carbohydrate), the authors found no difference in tumor growth or survival between both groups when saturated fat was used as the fat source [26]. Another study showed no difference in LNCaP tumor size between normal (6% fat) and high-fat (14% fat) diets [22]. Taken together, a number of studies involving subcutaneous human prostate cancer cell xenografts in immunodeficient mice suggested an association between HFD and xenograft growth, whereas several other studies showed no such relationship. The lack of standardization in terms of models and duration of specific diet feeding has remained problematic.

Given the variations in the genetic background of mouse strains, it is important to consider the importance of the immune system in tumor progression [59]. Several studies have investigated the impact of dietary fat on allografts using immunocompetent mice and mouse-derived prostate cancer cells [31,41,45,49,55]. Several groups have shown that a HFD significantly increased allograft tumor growth of TRAMP-derived prostate cancer cells, such as TRAMP-C1 and TRAMP-C2, in C57BL6 mice [31,41,55]. The study involving the largest number of allografts (low-fat; *n* = 40, high-fat; *n* = 134) revealed that mice receiving AIN-93M-high-fat diet had significantly heavier and significantly larger TRAMP-C2 allografts compared to those receiving AIN-93M, whereas no differences in prostate weight were observed among the groups [31]. This result suggests that TRAMP allografts derived from C57BL6 mice can be one of the promising allograft models when studying HFD-induced prostate cancer progression.

A unique study involving a peritoneal dissemination model established through intracorporeal injection of PC-3M-luc cells detected using the Xenogen IVIS™ system reported that a HFD increased tumor formation rates and total metastasis rates in the peritoneal organs [53].

In summary, given that most of the xenograft and allograft studies were performed using subcutaneous xenograft models, studies involving metastatic models and human patient-derived xenografts (PDXs) have been lacking. Although several studies using xenografts and allografts have shown that a HFD accelerated tumor growth, further validation is warranted.

### 2.2. TRAMP Mouse Models

Since its generation in 1996, the TRAMP mouse model has been one of the most widely used models in prostate cancer research [60]. This model represents a transgene comprising the minimal probasin promoter driving viral SV40 large-T and small-t antigens, which lead to prostate-specific inactivation of pRb and p53, specifically in the prostatic epithelium [61]. TRAMP mice develop prostatic intraepithelial neoplasia (PIN) by the time they are 6 weeks old; this progresses to high-grade PIN by the age of 12 weeks and poorly differentiated and invasive adenocarcinoma by the age of 24 weeks, with nearly 100% penetrance [61]. The impact of a HFD on the growth of TRAMP mouse tumors had been frequently evaluated [25,31,39,41,42,43,57,62]. Accordingly, Llaverias et al. showed that mice consuming a Western-type diet enriched in both fat and cholesterol had higher prostate tumor incidence and greater tumor burden compared to those fed a control chow diet [25]. After necropsy at 28 weeks, 33% of TRAMP mice fed a Western diet showed grossly evident spherical prostate tumors, whereas only 17% of TRAMP mice fed a chow diet exhibited the same [25]. In another study on TRAMP mice, Xu et al. revealed that the HFD group had significantly higher mortality than the normal diet group (23.81% and 7.14%, respectively, *p* = 0.035). Moreover, HFD-fed TRAMP mice had significantly higher tumor incidence at 20 weeks, as compared to the normal diet group (78.57% and 35.71%, *p* = 0.022, respectively) [43]. The same group also showed that HFD-fed mice suffered higher rates of extracapsular extension (20 weeks, 16.7% vs. 8.3%; 28 weeks, 66.7% vs. 50.0%, respectively) and distant metastasis (e.g., retroperitoneal lymph nodes or lung metastasis) (28 weeks, 41.7% vs. 25.0%, respectively) [62]. Bonorden et al. conducted a unique study involving the largest number of mice (*n* = 25 each) to assess the direct effect of diet and body weight on prostate tumors. TRAMP mice received low- and high-fat diets with the latter being divided into three groups: obesity-prone (the heaviest third), overweight (the middle third), and obesity-resistant (the lightest third). Accordingly, their results showed that body weight or diet had no effect of on either age at tumor detection, neuroendocrine status, or age at death [31]. Taken together, the impact of a HFD on tumor incidence and survival of TRAMP mice still remains controversial. The timing of diet change, selection of control diet, and diet ingredients may be important in establishing HFD-accelerated orthotopic prostate tumor models in TRAMP mice.

### 2.3. Other Genetically Engineered/Transgenic Mouse Models Targeting Oncogenes and Tumor Suppressor Genes

Several studies have investigated the effect of dietary fat on prostate cancer development and progression using genetically engineered mouse models (GEMMs) targeting oncogenes and tumor suppressor genes [19,29,30,36,40,47,48,54,56]. Designated Hi-myc uses a PB promoter coupled with a sequence of the ARR2 promoter, both of which lie upstream to the human c-Myc gene, in order to drive progression from mouse prostatic intraepithelial neoplasia (mPIN) to invasive adenocarcinoma [63]. Using this animal model, Kobayashi et al. showed that the HFD group (42 kcal% fat) had a greater number of invasive adenocarcinoma and a higher proliferative index in the PIN region compared to the LFD group (12 kcal% fat) [19]. Phosphatase And Tensin Homolog (Pten) alteration has been shown to be an early event in prostate cancer initiation and progression. Moreover, Pten-null mice that develop PIN have among the valuable animal models in prostate cancer research [64]. Kalaany et al. showed that 40% dietary restriction did not have any detectable effect on the extent or histological appearance of the PIN in Probasin-Cre; PTEN L/L prostate cancer models but significantly reduced tumor nodules in the lungs of K-RAS^LA2^; P53 LSL/WT lung adenocarcinoma models [23], suggesting that the Phosphoinositide 3-kinase (PI3K)/ protein kinase B (AKT) pathway is critical for diet-induced cancer progression. Conversely, a high-calorie diet (45 kcal% fat) promoted prostate cancer progression in genetically susceptible Pten haploinsufficient mice with increasing inflammatory response in the presence of enhanced insulin response to chronically elevated insulin levels [40]. Hayashi et al. demonstrated that mice receiving a HFD for 17 weeks starting from an age of 5 weeks had significantly higher prostate weights of than those receiving control [54]. Moreover, HFD-fed model mice had a significantly higher Ki67-positive cell to tumor cell ratio than control mice, while no marked difference in glandular structures was observed between the control diet (CD)-fed and HFD-fed model mice [54]. An interesting study involving the basal cell-specific Pten-null model using K14-Pten-mTmG mice showed that HFD intake promoted the initiation and progression of PIN lesions [48]. Although dietary fat could potentially be associated with prostate cancer development of Pten-null mice, the impact may not be extensive. Additionally, the evaluation of prostate pathology in GEMMs needs to be standardized according to the Consensus Report from the Bar Harbor Meeting of the Mouse Models of Human Cancer Consortium Prostate Pathology Committee for accurate comparison among different studies [65].

With regard to other GEMMs, PTP1B (PTPN1), an androgen-regulated phosphatase, acts as a HFD-dependent tumor suppressor in prostate cancer driven by the absence of Pten, such as in the Pten-/-Ptpn1-/- mice model [47]. Deficiency in RXRα (a unique and important member of the nuclear receptor superfamily) in the prostates of mice receiving a new Western-style diet resulted in higher rates of mPIN and prostate cancer [30]. Pommier et al. showed that mice with double knockout of Liver X receptors (LXRa and LXRb), which belong to the nuclear receptor superfamily and are central mediators of cholesterol homeostasis, developed PIN under a diet high in cholesterol [36].

Reports regarding HFD-induced metastatic models using GEMMs have been rare. Chen et al. showed that among mice with Pten deletion and a double deletion of Pten and Promyelocytic Leukemia (PML), a suppressor of pp1α-dependent activation of MAPK signaling, those receiving a lard-based HFD displayed lymph node metastasis and lung metastasis, whereas those receiving a chow diet exhibited limited metastases [56].

Taken together, GEMM studies showed that a HFD enhanced tumor growth through the modulation of several genes, including those related to PTEN. Studies that assess the impact of a HFD using more aggressive, metastatic GEMMs while considering the effect of dual and/or multiple genes may be intriguing.

### 2.4. Others

Several studies have evaluated the proliferation of prostate cancer cell lines cultured with serum from mice and humans under different diet conditions [13,27]. Two mice studies proposed that a HFD serum enhanced cell proliferation of LAPC-4 and PC-3/DU145 cells in CB17 SCID and Balb-c/nu/nu mice, respectively [15,52]. With regard to in vitro studies using human sera, Barnard et al. assessed the growth of LNCaP cells cultured with healthy volunteer serum according to dietary fat and exercise condition [14]. Accordingly, they found that an LFD with exercise inhibited cell growth. Subsequently, after evaluating the growth of LNCaP cells cultured with sera from patients with prostate cancer receiving a low-fat, high-fiber, soy-protein supplement diet or Western diet for 4 weeks, Aronson et al. showed that the LFD induced changes in serum fatty acid levels with decreased LNCaP cancer cell growth [27]. In an interesting study by Lo et al., PDX models of prostate cancer cells implanted into the renal capsule of SCID mice were developed [44]. Histological analysis of the PDXs showed no differences in tumor pathology; PSA, androgen receptor, and homeobox protein Nkx-3.1 expression; or proliferation index between HFD- and LFD-fed mice. Furthermore, they also evaluated the impact of co-grafting human periprostatic adipose tissue (PPAT) with prostate cancer in PDX grafts. After harvesting the PDX tissues 10 weeks after grafting, histological analysis revealed no evidence of enhanced tumorigenesis with PPAT compared to prostate cancer grafts alone. It would be intriguing to assess the effects of a HFD on PDXs with a more aggressive prostate cancer phenotype obtained from metastatic disease and co-grafting this with PPAT from patients with severe obesity considering that the aforementioned model was established using tissues from patients with localized prostate cancer treated with surgery.

## 3. Differences in Diets

A number of studies have tried to assess the impact of a fat-enriched diet on prostate cancer development and progression. However, the dietary interventions had not been standardized, while distinct variations in phenotype had been observed among different studies using distinct HFD components. Differences in dietary components among research models had also been reported to affect the distinct effect of diet-induced metabolic disorders [66]. Therefore, in addition to the models used, the type of diet remains essential for studies to delineate diet-induced carcinogenesis.

### 3.1. Direct Comparison between Two Different Diets Including the High-Fat Diet and Another Diet

While conclusions have been frequently drawn from comparisons between a defined HFD and chow, specific details regarding the control diet are often lacking. Many studies have utilized a chow diet as the control treatment [25,26,49,50,56]. Regular chow is composed of agricultural byproducts, such as ground wheat, corn or oats, alfalfa, and soybean meals; a protein source, such as fish; and vegetable oil; it is supplemented with minerals and vitamins. Thus, chow can be considered a high-fiber diet containing complex carbohydrates with fats from various vegetable sources. Chow is inexpensive to manufacture and palatable for rodents. In contrast, defined HFDs consist of amino acid-supplemented casein, cornstarch, maltodextrin or sucrose, and soybean oil or lard and are supplemented with minerals and vitamins. Fiber is often provided by cellulose. Chow and defined diets may exert significant separate and independent unintended effects on the measured phenotypes in any research protocol [66]. In sum, multiple limitations may affect the results of the target groups.

A direct comparison between two different diets including the HFD, has been used extensively to understand the role of diet on prostate cancer development and progression [12,15,27,30,43,47,51,58]. Most of the studies showed that HFD-fed mice had greater body weight compared to controls, which leads one to consider whether diet has a direct or indirect (obesity-induced) effect on cancer development and progression. Although majority of the previous studies proposed an association between dietary fat and prostate cancer development/progression, several limitations need to be considered. First, a multitude of proportions per calories of fat have been observed with relative fat fractions ranging between 14% and 84% energy as fat. We need to consider the fact that the higher proportions of fat used in animal studies cannot be used in human diets. Second, we need to be careful about being misled by ignoring the impact of fat components, the control diet, and other elements in each diet. For instance, Lloyd et al. showed no difference in the growth and survival of LAPC-4 xenografts between SCID mice receiving a Western-style diet, including 40% kcal fat, and those fed an LFD (12% kcal) [26]. In this study, the fat consisted of 19% lard, 19% milk fat, and 1.9% corn oil. Conversely, another study demonstrated that HFD-fed SCID mice (42% kcal) had significantly faster LAPC-4 tumor xenograft growth and higher PSA levels compared to LFD-fed mice (12% kcal) [15]. In this study, the HFD was composed of corn oil. These lines of evidence suggest that different effects have been observed despite having similar percentages of fat components. Finally, publication bias should be taken into account for a comprehensive understanding, because negative data tend to remain unpublished.

### 3.2. Comparison of the Impact of a High-Fat Diet using Multiple Diets

A direct comparison between multiple diets using animal models is one method of identifying the diet having the most effect on tumor growth. In our previous study, LNCaP xenograft tumor growth in Balb/c-nu/nu mice were evaluated among three groups receiving a HFD (59.9 kcal% fat), Western-style diet (WD: 41.2 kcal% fat), and high carbohydrate diet (HCD: 9.5 kcal% fat) [33]. Accordingly, our results showed that the HFD group had significantly higher LNCaP xenograft tumor growth than the HCD and WD groups. In general, a ketogenic diet, which contains extremely high fat, is toxic to cancer [28]. Accordingly, the systematic review by Khodadadi et al. demonstrated that a ketogenic diet can potentially inhibit malignant cell growth and increase survival time [67]. Moreover, studies comparing the tumor growth and survival of LAPC-4 xenografts in SCID mice demonstrated that mice receiving a no-carbohydrate ketogenic diet (NCKD: 83% fat, 0% carbohydrate, 17% protein) had smaller tumors and higher survival than those receiving a low-fat/high-carbohydrate diet (LFD: 12% fat, 71% carbohydrate, 17% protein) or a high-fat/moderate carbohydrate diet (MCD: 40% fat, 43% carbohydrate, 17% protein) [20]. Another study also investigated the differences between three diets, namely a NCKD (84% fat–0% carbohydrate–16% protein kcal), 10% carbohydrate diet (74% fat–10% carbohydrate–16% protein kcal), and 20% carbohydrate diet (64% fat–20% carbohydrate–16% protein kcal), with results showing significantly larger tumors in the 10% carbohydrate group but no difference in survival [28]. These lines of evidence suggested that extremely high fat percentages have a potential to exert an opposite effect on prostate cancer development and progression. Therefore, the proportion of total fat intake remains important.

One study using a Western-type diet (16% protein, 40% fat, 44% carbohydrate) evaluated the impact of seven diets: Group 1, ad libitum 7 days/week; Group 2, fasted 1 day/week and ad libitum 6 days/week; Group 3, fasted 1 day/week and fed 6 days/week via paired feeding to maintain isocaloric conditions similar to that in Group 1; Group 4, 14% calorie restriction (CR) 7 days/week; Group 5, fasted 2 days/week and ad libitum 5 days/week; Group 6, fasted 2 day/ week and fed 5 days/week via paired feeding to maintain isocaloric conditions similar to that in Group 1; and Group 7, 28% CR 7 days/week [24]. Accordingly, some of the groups did not exhibit trends toward tumor shrinkage and improved survival, although Groups 6 and 7 had lower lean body mass than Group 1 in a two-way comparison. The study implicated that intermittent calorie restriction via fasting with a Western-style diet had no impact on prostate cancer progression, despite the effect on body weight.

### 3.3. Specific Components of Fat

Each dietary fat has diverse physiological effects according to the different types and distributions of dietary fat components. Therefore, important relationships between specific types of dietary fat intake and prostate cancer development may be missed by merely evaluating the effect of total fat intake [68]. Fatty acids are classified based on whether or not the fatty acid carbon chain contains no double bond (saturated fatty acids (SFA)), one double bond (monounsaturated fatty acids (MUFA)), and more than one double bond (polyunsaturated fatty acids (PUFA)), as well as the configuration of the double bonds (*cis* or *trans*). In addition, PUFA are often classified based on the position of the first double bond from the fatty acid methyl terminus, creating omega-3 and -6 fatty acids. The primary sources for SFA, MUFA, and PUFA include animal fats such as lard and beef tallow, animal and certain vegetable fats such as olive oil, and vegetable oil such as corn and fish oils, respectively [66]. Corn oil and most vegetable oils contain omega-6 PUFA, whereas fish oils are high in omega-3 PUFA [69].

In general, a number of previous studies made use of a lard-based HFD, which is rich in SFA. Studies in human subjects have shown that SFA are more oncogenic than PUFA [70]. Moreover, several studies have shown that cancerous tissues exhibited elevated SFA and MUFA compared to adjacent normal tissues [71,72]. Mice receiving lard oil had been reported to have enhanced Toll-like receptor (TLR) activation and white adipose tissue inflammation, as well as reduced insulin sensitivity, compared to those receiving fish oil [69], suggesting that a diet rich in SFA accelerated metabolic inflammation. In general, MUFA, such as oleic acid and olive oil, are more likely to prevent or decrease the risk of carcinogenesis in other solid cancers, including breast and colon cancers [73]. Phenolic compounds, which prevent free radical-initiated peroxidation and regulate cancer-related oncogenes, have been considered to be associated with MUFA-induced chemoprevention [73]. Omega-3 and -6 PUFA are essential fatty acids that mammals can neither synthesize nor de novo interconvert, suggesting that they have to be obtained from the diet [18]. From an evolutionary standpoint, the human diet has had a 1:1 ratio of omega-6-to-omega-3 PUFA [74]. Over the past two centuries, however, this ratio has increased to nearly 10:1 due primarily to the increased use of vegetable oils in Western diets [8,45]. In general, the high consumption of omega-6 fatty acids leads to inflammation and cellular growth through the conversion of arachidonic acid (an omega-6 fatty acid) to hydroxyeicosatetraenoic and epoxyeicosatrienoic acids by cytochrome P450 oxygenases [75]. In contrast, omega-3 induces anti-inflammatory, pro-apoptotic, anti-proliferative, and anti-angiogenic pathways, providing antitumor effects against prostate cancer [76]. Fish oil, which contains omega-3 fatty acids, does not cause obesity because of peroxidization [77] and induces the activation of peroxisome proliferator-activated receptor alpha. These lines of evidence suggest that omega-3 and -6 PUFA have different effects on diet- and obesity-induced prostate cancer development and progression. Three studies had reported on the difference in tumor growth between diets rich in omega-3 and -6 [18,29,45]. Accordingly, mice fed a high omega-3 diet had significantly lesser prostate weight gain than those fed a high omega-6 diet. Moreover, half of the mice fed a high omega-3 diet developed invasive carcinoma, whereas 80% of mice fed a high omega-6 diet had invasive carcinoma [18]. The second study revealed that fish oil slowed the progression of tumorigenesis in dorsolateral prostate C3 (1) tag transgenic mice [29]. The last study, which established MycCaP allografts in immunocompetent FVB mice, found that the ω-3 group had significantly smaller tumor volumes than the ω-6 group [45]. All three different models successfully confirmed that omega-3 inhibited tumor growth, which suggests the promising inhibitory effects of omega-3 fatty acid against prostate tumors.

Cholesterol, an organic compound, is a key component of membrane signaling microdomains. In humans, cholesterol can be either obtained from diet or synthesized de novo in the liver. Animal studies using the cholesterol uptake inhibitor ezetimibe for prostate cancer chemoprevention showed that lowering serum cholesterol level slows tumor growth and decreases angiogenesis and intratumoral androgens [78]. Pommier et al. demonstrated that a high-cholesterol diet induced proliferation in LXR mutant mouse prostate [36]. In a clinical setting, the meta-analysis performed by Bonovas et al. was the only study to find a significantly reduced incidence of advanced prostate cancer in subjects who were prescribed statins; however, no relationship between statin use and overall prostate cancer risk was demonstrated in other studies [79]. The observational study by Murtola et al. reported a dose-dependent, significant inverse association between overall prostate cancer incidence and statin use, with the strongest inverse association for early-stage prostate cancer [80]. However, clinical evidence on the protective effect of cholesterol-lowering drugs for prostate cancer chemoprevention is still weak and inconsistent; therefore, we are unable to draw a firm conclusion based on these results.

Finally, care should be taken when establishing how much of a role other nutrients contained in experimental diets have and the actual consumption of diets in each mouse given that the proportion of other ingredients changes when the percentage of fat components is modulated. 

## 4. Potential Mechanisms

Previous studies have proposed several mechanisms in order to explain the possible association between dietary fat and prostate cancer development/progression. Accordingly, growth factor signaling, lipid accumulation, inflammation, and endocrine modulation had been hypothesized to be associated with HFD-induced prostate cancer development and/or progression (Figure 1). Certainly, a more thorough understanding of the possible association between dietary fat and prostate cancer risk requires further inquiry.

### 4.1. Growth Factor Signaling

Obesity and hyperinsulinemia have been associated with increased amounts of circulating bioactive insulin-like growth factor-I (IGF-I), a growth factor determined to play a pathogenic role in many cancers [81]. Barnard et al. demonstrated that dietary fat reduction combined with a regular exercise intervention in men decreased serum IGF-I and increased serum IGFBP-1 levels, resulting in decreased growth of LNCaP human prostate cancer cells cultured in media containing volunteer serum [14]. The same group showed that LFD-fed mice had significantly slower tumor growth rates, lower levels of serum insulin, tumor IGF-I mRNA expression, and tumor IGFBP-2 immunostaining, and higher levels of serum IGFBP-1, which indicated that IGF-I signaling modulated fat-induced tumor growth in LAPC-4 xenografts [15]. We had previously demonstrated that IGF-I receptor (IGF-IR) mRNA levels were strikingly elevated in HFD-accelerated LNCaP xenografts and that the group having the lowest IGF-IR immunoreactivity tended to have the lowest body mass index in both human normal and prostate cancer epithelia [16]. Kobayashi et al. showed that an LFD reduced the development of prostate cancer in Hi-Myc mouse transgenic model with the suppression of the IGF-AKT pathway, which leads to higher serum IGFBP-1 levels, reduced serum mitogenicity, and lower AKT, GSK3beta, and S6K activities [19].

Several studies have demonstrated that hyperactivation of PI3K-AKT, which is one of the downstream targets for IGF-I signaling, desensitizes tumors to dietary modulations, including calorie restriction and a HFD [23,47]. The PI3K/AKT pathway is naturally inhibited by Pten, which is one of the most frequently lost or mutated tumor suppressor genes in prostate cancer [56]. Partial loss of *PTEN* is observed in 70% of localized prostate cancer, while complete loss thereof is associated with metastatic castration-resistant prostate cancer [56]. *PTEN* inactivation also induces aberrant activation of the PI3K/AKT pathway. As previously mentioned, conditional PTEN knockout produces indolent tumors in mouse prostates. One study that assessed the impact of diet restriction revealed that it does not affect a PTEN-null mouse model of prostate cancer but significantly decreases tumor burden in a mouse model of lung cancer lacking constitutive PI3K signaling, which suggests that PI3K signaling is strongly associated with diet-induced cancer progression [23]. Another study involving a GEMM mouse model showed that the loss of both PTEN and the protein tyrosine phosphatase Pypn1, a negative regulator of IGF-IR, enabled the development of a highly invasive prostate tumor, whereas PTEN deficiency alone resulted in tumors that were unresponsive to HFD [47]. Collectively, mechanisms involving PTEN and other related genes may have a higher impact on diet-induced prostate cancer development and progression.

Many other studies have proposed that IGF-I/PI3K/AKT signaling has an impact on diet-induced prostate cancer development and progression [21,28,43]. Therefore, IGF-I/PI3K/AKT signaling has been one of the promising pathways related to HFD-induced prostate cancer development and progression. To determine the impact of treatment, the additive effect of IGF-1R inhibition using IGF-IR blockade antibody on 22Rv1 subcutaneous xenografts in SCID mice receiving a HFD (43.3%) or LFD (12.4%) had been investigated [32]. Accordingly, the results showed that the LFD + IGF-1R-Ab group had a significantly smaller mean tumor volume compared to the HF group at day 14 of the intervention. However, no significant difference in final tumor volumes or final tumor weights had bene observed between the four treatment groups. Therefore, the therapeutic effect of IGF-I pathway inhibition remains unknown.

Diet-induced hyperinsulinemia has been shown to accelerate tumor growth in different prostate cancer xenograft models [17,20]. A large prospective survival analysis reported that higher serum C-peptide concentrations, a surrogate of insulin levels, were associated with increased prostate cancer-specific mortality [82]. Insulin and IGF-I are closely related hormones that act on specific tyrosine kinase receptors and elicit the activation of a cascade of intracellular proteins leading to the regulation of gene expression, protein synthesis, cell proliferation or death, and glucose and lipid metabolism. High insulin levels, as well as insulin receptor and IGF-I/IGF-IR axis activation, have been known to be associated with obesity induced cancer progression [83]. Regarding the impact of insulin levels on diet-induced prostate cancer growth, one study involving LAPC-4 xenografts in SCID mice receiving three different diets, NCKD (84% fat), 10% carbohydrate diet (74% fat), or 20% carbohydrate diet (64% fat), proposed that mice receiving a 10% carbohydrate diet had larger tumors than the other groups despite mice receiving a 20% carbohydrate diet having the lowest insulin levels [28]. As such, future studies need to elucidate the relationship between insulin levels and diet-induced prostate cancer carcinogenesis.

In addition, effects of different fat sources on the IGF/insulin axis have rarely been discussed and studied. It would be intriguing to know the varying impacts specific fats have on HFD-induced prostate cancer development and progression through growth factor signaling.

### 4.2. Lipid Accumulation

The changes in endogenously synthesized/exogenous lipid profiles and related enzymes have been linked to prostate cancer development and progression. Accordingly, Freedland et al. showed that mice with LAPC-4 xenografts receiving a NCKD diet had low hepatic fatty infiltration, which resulted in reduced tumor growth and longer survival [20]. Genome-wide gene expression analysis showed that the lipogenic gene ELOVL7, which possibly codes a long-chain fatty acid elongase, was overexpressed in clinical prostate cancer and regulated by SREBP1. Moreover, a HFD had been found to promote the growth of in vivo tumors of ELOLV7-expresssed prostate cancer [22]. In the aggressive and metastatic tumor progression observed in TRAMP mice receiving a Western-style diet, Llaveries et al. showed that the Western-style diet increased both the expression of the high density lipoprotein receptor SR-BI and angiogenesis [25]. Fatty acid synthase (FASN) is a cytosolic metabolic enzyme that catalyzes de novo fatty acid synthesis. Our previous study found that serum FASN levels were significantly lower and were inversely correlated with tumor volume in LNCaP xenograft mice receiving HFD [46].

A recent study has suggested that a Western-style HFD promotes metastatic prostate cancer through a prometastatic lipogenic program alteration [56]. In this study, the conditional inactivation of Pml (a suppressor of pp1α-dependent activation of MAPK signaling) in mouse prostates changed indolent PTEN-null prostate tumors into lethal metastatic tumors with MAPK reactivation, subsequent hyperactivation of an aberrant SREBP, and a lipidomic profile alteration. Pten-/-, Pml-/- mice receiving a chow diet displayed limited lymph node metastasis. However, most mice receiving a HFD developed lymph node metastasis, while half of them had lung metastasis. Moreover, Oil Red O staining showed that the tumors in mice receiving a HFD had higher lipid accumulation compared to those in mice receiving a chow diet. Sterol responsive element binding proteins (SREBPs) have been found to be a key regulator of lipogenic genes [56]. Studies have shown that HFD feeding stimulates SREBP expression, subsequent expression of genes encoding lipogenic enzymes, and lipid accumulation in nonadipose tissues [56,84]. Therefore, Pml and SREBP-dependent lipogenic alterations may be associated with HFD-enhanced prostate cancer progression.

Although de novo lipogenesis has emerged as an important player in prostate cancer, the impact of exogenous dietary fat on intraprostatic lipid profiles and activity of lipogenic enzymes remains largely unknown. Future studies are required to elucidate the mechanisms for endogenous lipid alterations and exogenous fat accumulation on dietary fat-induced prostate cancer development and progression.

### 4.3. Inflammation

Inflammations have been shown to promote the development and progression of prostate cancer [85]. A HFD with consequent obesity causes adipose tissue inflammation and cytokine secretion [86]. Accordingly, Liu et al. demonstrated that Pten +/- mice receiving a high-calorie diet exhibited neoplastic progression with stromal infiltration of inflammatory cells, such as macrophages, T cells, and inflammatory monocytes, into the prostates [40]. Additionally, the increased inflammatory response to a high-calorie diet was supported by the elevation in the expression of CD3, CD45, FoxP3, MCP-1, IL-6, and TNF alpha. Microarray analysis using TRAMP mice models showed that HFD feeding increased serum levels of MCP-1, MCP-5, TIMP-1, IL-16, CCL12, CXCL1, CXCL10, and CXCL13 [41]. Similar results were observed in the sera of TRAMP-C2 allograft models. Zhang et al. demonstrated that adipose stromal cell recruitment to tumors of RM1 mouse prostate cancer xenografts via CXCL1 and CXCL8 chemokines promoted prostate cancer progression [49]. Another study showed that MycCaP xenografted immunocompetent FVB mice receiving a diet rich in omega-3 exhibited tumor suppression, as well as lower gene expression of markers for M1 and M2 macrophages, associated cytokines (IL-6, TNF alpha, and IL-10), and the chemokine CCL-2. Hayashi et al. showed that a HFD increased the prostate weight and percentage of Ki67-positive MDSCs, as well as the M2/M1 macrophage ratio, in HFD-fed model mice with a higher serum IL-6 levels [54]. Furthermore, celecoxib suppressed tumor growth in HFD-fed but not CD-fed model mice, which suggested that HFD-induced tumor growth was associated with local inflammation. Taken together, tumor-infiltrating macrophages may perhaps be a key factor in HFD-induced prostate cancer progression [54]. One of our previous studies had demonstrated that the MCP-1/CCR2 pathway, a key regulator of macrophage infiltration, was highly associated with HFD-induced LNCaP xenograft tumor growth, supporting the results presented herein [33]. We also found that the expression of macrophage inhibitory cytokine 1 (MIC-1), a divergent member of the transforming growth factor beta, was stimulated by palmitic acid in vitro, while mice receiving a HFD containing high amounts of palmitic acid had LNCaP had significantly greater xenograft tumor growth, serum MIC1 levels, and fatty acid levels in xenograft tumors than those receiving an LFD in vivo [37]. Such lines of evidence suggesting the association between cytokines and tumor–macrophage interaction support the notion that tumor-associated macrophages play a role in HFD-induced prostate cancer development and progression. Another mouse xenograft experiment concluded that HFD enhanced prostate cancer metastasis and invasiveness through FABP4 and interleukin-8 upregulation [53]. FABP4, an abundant protein in adipocytes that is influenced by a HFD or obesity, may enhance prostate cancer progression and invasiveness by upregulating matrix metalloproteinases and cytokine production in the prostate cancer stromal microenvironment [53]. In one study demonstrating tumor growth decline among Pten KO mice receiving omega-3 fatty acid, the group with an omega-3-enriched diet exhibited a reduction in CD3+ lymphocyte levels and tumor microvessel density [18]. These lines of evidence suggest that a stromal microenvironment, including infiltration of immune cells, is associated with dietary-fat induced prostate cancer carcinogenesis. Moreover, other cytokines and chemokines, including TWEAK, (CCL)3, CCL4, and CCL5, had been found to be potentially associated with HFD-induced prostate cancer progression according to previous literatures [41,57,87].

The role of adipocyte function on HFD feeding was evaluated in a recent study [50]. Accordingly, Cav-1 secretion from fat tissue of HFD-fed mice was increased, while hypertrophied adipocytes were responsible for enhanced Cav-1 secretion in obese mice. Furthermore, secreted Cav-1 was taken up by the preadipocytes and LNCaP cells. The impact of hypertrophied adipocyte-induced Cav-1 secretion on prostate cancer progression and diet- and/or obesity-modulated adipose function could be associated with prostate inflammation and prostate carcinogenesis. Moreover, adipose tissues are known to enhance cancer progression via several underlying mechanisms, such as aromatization of adrenal androgens to estrogens in the adipocyte and deregulation of the expression and secretion of the adipokines [88,89]. Therefore, it is worthwhile to assess the role of quantitative and qualitative modulation of adipose tissues on prostate cancer progression using preclinical models. 

In summary, the interaction among systematic and/or adipose secreted cytokines, inflammation, and immune cell infiltration into tumors may have a promising mechanistic role in HFD-induced prostate cancer development and progression.

### 4.4. Endocrine Modulation

Considerable epidemiological evidence has shown that fat-containing diets may increase the risk of certain hormone-dependent conditions in men via its effects on hormone metabolism [90]. Hormonal modulation has been one of the proposed mechanisms associated with diet- and/or obesity-induced prostate cancer carcinogenesis [8,9] given that sex hormones play a key role during normal and cancerous prostate growth and development. Two transgenic mouse studies showed that omega-3 fatty acids slowed prostate tumor growth through the modulation of sex steroid pathways. In the C3 (1) Tag transgenic mice study, the lowering testosterone, estradiol, and androgen receptor levels by the action of omega-3 fatty acids promoted apoptosis and suppressed prostate epithelial cell proliferation [29]. Another study demonstrated that omega-3 PUFA treatment slowed castration-resistant tumor growth and accelerated androgen receptor protein degradation [34]. A recent study showed that although serum cholesterol reduction did not significantly affect the rate of adenocarcinoma development in the PTEN-null transgenic mouse model of prostate cancer [78], it lowered intraprostatic androgens and slowed tumor growth. These results suggest that fat-containing diets, especially those that modulate of omega-3 fatty acid content, may potentially modulate intraprostatic hormonal status associated with cancerous tumor growth and progression. In the TRAMP-C1 allograft study, HFD increased tumor growth and serum estradiol levels [55]. The study also showed that intratumoral C-terminal-binding protein 1 (CtBP1) controls the transcription of aromatase (CYP19A), a key enzyme that converts androgens to estrogens, and was overexpressed with increased TRAMP-C1 allograft tumor growth in mice receiving a HFD. In another study, Moiola et al. found that mice with CtBP1-depleted PC-3 xenografts developed significantly smaller tumors than those inoculated with PC-3 control cells [38]. These results suggest that CtBP1 have the potential to be a key transcriptional factor associated with intratumoral hormonal modulation and HFD-induced prostate cancer growth.

### 4.5. Others

Using microarray analysis, our group had previously showed that several mRNAs and miRNAs become altered in HFD-induced LNCaP xenografts [16,52]. Therefore, complex mechanisms, including candidate pathways mentioned previously, may be considered to contribute to fat-diet induced prostate cancer development and progression. Nara et al. demonstrated that miR-130a was attenuated in HFD-induced prostate cancer progression with MET overexpression in vitro and in vivo and that cytoplasmic MET in prostate cancer tissues was overexpressed in patients with higher body mass index [52]. Kim et al. found that a HFD not only accelerated Src-induced prostate tumorigenesis, but also compromised the inhibitory effect of the anticancer drug dasatinib on Src kinase oncogenic potential in vivo [51]. Finally, the association between diet-induced prostate cancer progression and several pathways, including oxidative stress [39], epithelial–mesenchymal transition [40], and basal/luminal differentiation [48], have been proposed in previous studies.

## 5. Concluding Remarks

Over recent years, the molecular mechanisms behind HFD-induced prostate cancer development and progression have been studied using pre-clinical models. Although several lines of evidence have proposed its relationship with potential mechanisms, such as growth factor signaling, lipid accumulation, inflammation, and endocrine modulation, the current data still remains inconclusive. In addition, the studies presented herein have used various types of models and diet sources, suggesting the need for increases vigilance when communicating and interpreting information. Therefore, it is important to consider the predictability and limitation of each preclinical model when translating experimental results into clinical practice. Although information from pre-clinical models remain important for deeper understanding and exploration of novel treatment targets, further studies are needed to validate the impact of dietary fat and obesity on prostate cancer development and progression.

## Figures and Tables

**Figure 1 jcm-08-00597-f001:**
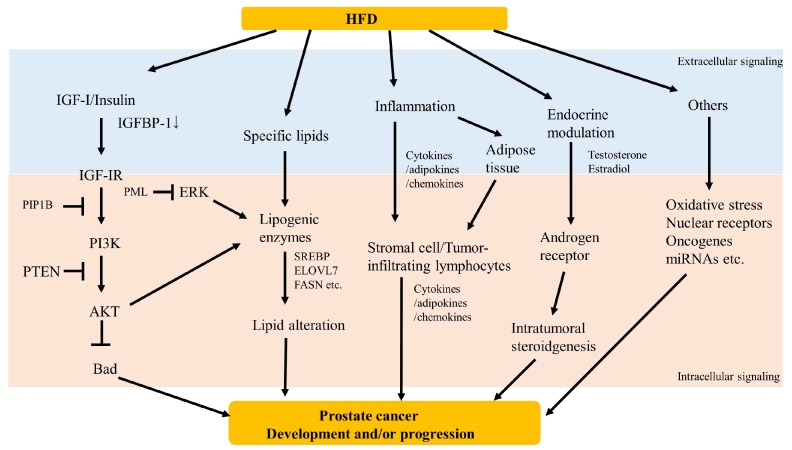
Scheme of potential mechanisms underlying high-fat diet induced prostate cancer development and/or progression.

**Table 1 jcm-08-00597-t001:** Summary of preclinical models on dietary-fat induced prostate cancer development and progression.

Authors	Years	Animal Models	Tumors	Diet Summary	End Point	Summary of the Results
Wang [11]	1995	Nude mice	LNCaP	40.5%, 30.8%, 21.2%, 11.6%, or 2.3l% fat	Tumor growth rates, tumor weights, ratios of final tumor weights to animal weights, PSA	Groups that continued to receive a 40.5l% fat diet were substantially greater tumor growth rates, final tumor weights, and ratios of final tumor weights to animal weights than those whose diets were changed to 2.3 kcal%, 11.6 kcal%, or 21.2 kcal% fat.
Connoly [12]	1997	Nude mice	a) DU145 subcutaneous xenograft, b) DU145 into prostate	a) 18:2 ω-6-rich vs. 18:3 ω-3-rich vs. 20:5 and 22:6 ω-3-rich, b) ω-6-rich vs. a LF	Tumor growth	a) 18:2 ω-6-rich vs. 18:3 ω-3-rich mice were similar; a 30% reduction in tumor growth was observed in the 20:5 and 22:6 ω-3-rich groups. b) The mean tumor weight in the ω-6-rich group was twice that in the low-fat group.
Ngo [13]	2002		LNCaP cultured with human serum	Before and after residential diet and exercise	Cell growth, apoptosis, necrosis	Serum-stimulated LNCaP cell growth was reduced by 30% in post-11-day serum and by 44% in long-term serum relative to baseline. LNCaP cells incubated with post-diet and exercise serum showed higher apoptosis/necrosis, compared to baseline.
Barnard [14]	2003		LNCaP cultured with human serum	Volunteer serum (control, LF and exercise, exercise alone)	Cell growth	Both the LF/exercise and exercise alone groups had reduced LNCaP cell growth compared to control.
Ngo [15]	2003	CB17 SCID	a) LAPC-4 xenograft, b) LAPC-4 culture with 10% mouse serum	HFD (42%) vs. LFD (12%)	a) tumor growth, PSA, b) cell growth	LFD mice had significantly slower tumor growth rates and lower serum PSA levels compared to HFD mice. LAPC-4 cells cultured in vitro with media containing serum from LFD mice demonstrated slower growth than LAPC-4 cells cultured in media containing HFD mice serum.
Ngo [16]	2004	CB17 SCID	LAPC-4 xeograft	HFD (42%) vs LFD (12%)	Tumor growth, survival	Tumor latency and mouse survival were significantly longer in the LFD castration versus HFD castration group.
Venkateswaran [17]	2007	Swiss nu/nu	LNCaP xenograft	HC + HFD vs. LC + HFD	Tumor growth	Mice on the HC–HFD diet experienced increased tumor growth.
Berquin [18]	2007	Prostate-specific Pten deletion mouse		High ω-6 vs. ω-3 diet	Prostate weight, rate of invasive carcinoma	Prostate weight was significantly lower in mice fed high ω-3; half of the mice fed ω-3 develop invasive carcinoma, whereas 80% of the mice fed high ω-6 diet had invasive carcinoma.
Kobayashi [19]	2008	Prostate specific High-Myc transgenic mouse	a)LNCaP, b)MycCap with mice serum	HFD (42%) vs LFD (12%)	Rate of mPIN and cancer incidence	The number of mice that developed invasive adenocarcinoma at 7 months was 27 % less in the LFD group (12/28) compared to the HFD group (23/33, *p* = 0.04). Epithelial cells in PIN lesions in the LFD group had a significantly lower proliferative index compared to epithelial cells in the HFD group (21.7% vs. 28.9%, *p* < 0.05).
Freedland [20]	2008	SCID	LAPC-4 xenograft	NCKD (84% fat) vs. LFD (12% fat) vs. WD (40% fat)	Tumor growth, survival	NCKD mice tumor volumes were 33% smaller than WD mice (rank-sum, *p* = 0.009). No differences in tumor volume were observed between LFD and NCKD mice with the latter having the longest survival.
Narita [16]	2008	BALB/c-nu/nu	LNCap xenograft	HF (56.7%) vs. LF (10.2%)	Tumor volume, PSA	Tumor volume and serum PSA levels were significantly higher in the HFD group than in the LFD group.
Mavropoulos [21]	2009	SCID	LNCaP xenograft	NCKD (83% fat) vs. LFD (12% fat) vs. WD (40% fat)	Tumor growth, survival	Tumor volumes in the WD group remained significantly larger than tumor volumes in the LFD and NCKD groups. Survival was significantly prolonged for the LF (hazard ratio, 0.49; 95% confidence interval, 0.29–0.79; *p* = 0.004) and NCKD groups (hazard ratio, 0.59; 95% confidence interval, 0.37–0.93; *p* = 0.02).
Tamura [22]	2009	Nude mice	LNCaP xenograft	HFD (14%) vs LFD (6%)	Tumor growth	LNCaP-Mock cells did not reveal any significantgrowth promotion by breeding with HFD. HFD breeding significantly promoted the growth of LNCaP-ELOVL7-1 cells in vivo (*p* = 0.0081).
Kalaany [23]	2009	Prostate-specific Pten deletion mouse		Ad libitum vs. CR	Percentage of proliferation and apoptosis	CR did not affect a PTEN-null mouse model of prostate cancer but significantly decreased tumor burden in a mouse model of lung cancer lacking constitutive PI3K signaling.
Bushemeyer [24]	2010	SCID	LAPC-4 xenograft	7 types of diet	Tumor growth, survival	No significant differences in tumor volume were observed among the various groups at any time point. Overall, the treatment group was not significantly related to survival.
Llaverias [25]	2010	TRAMP mouse		WD (21.2%) vs. chow (4.5%)	Prostate tumor incidence and progression	TRAMP mice fed a WD were shown to develop larger tumors compared to mice fed a chow diet. 67% (6 of 9 mice) of TRAMP mice fed a WD exhibited at least one metastatic focus, whereas 43% (3 of 7 mice) of mice fed a chow diet exhibited the same.
Lloyd [26]	2010	SCID	LAPC-4 xenograft	WD (40%) vs. chow (12%)	Tumor growth, survival	No difference in tumor growth or survival between chow and WD was observed.
Aronson [27]	2010		LNCaP cultured with human serum	PCa men with LF, high-fiber, soy protein-supplemented diet or WD for 4 weeks	Cell growth	LF, high-fiber, soy protein-supplement diet decreased LNCaP cancer cell growth.
Masko [28]	2010	SCID CB17	LAPC-4 xenograft	NCKD (84% fat), 10% carbohydrate diet (74% fat), or 20% carbohydrate diet (64% fat).	Tumor volume, PSA, survival	Tumors were significantly larger in the 10% carbohydrate group on days 52 and 59 (*p* < 0.05) and at no other point during the study. Diet did not affect survival (*p* = 0.34).
Akinsete [29]	2012	C3 (1) Tag transgenic mouse		High ω-6 vs. ω-3 diet	Tumor progression, apoptosis	Slower progression of tumorigenesis and enhanced apoptosis was observed in dorsalateral prostate of high ω-3 diet mice than in high ω-6 diet mice.
Mao [30]	2012	Homozygous prostate-specific RXRα knockout mouse		NWD (higher fat content, reduced calcium, vitamin D, and fiber) or AIN-76A		A significant joint effect of NWD and RXRα status in developing mPIN, but interaction was not significant owing to the small sample size.
Bonorden [31]	2012	a) TRAMP mouse, b) C57/BL6	b) TRAMP-C2 allograft	LFD (AIN-93M) vs. AIN-93M-HFD (33%)	a) tumor differentiation, percentage of metastasis, b) tumor weight and volume	No difference in the prostates of TRAMP mice. TRAMP-C2 cells grew faster when the mice were fed a HFD.
Konijeti [32]	2012	SCID	22Rv1	HFD (43.3%) + saline, HFD + IGF-1R-Ab, LFD (12.4%) + saline, LFD + IGF-1R-Ab	Tumor volume	No significant differences in final tumor volumes or final tumor weights were observed between the treatment groups. At day 14 of the intervention, the mean tumor volume was significantly lower in the LFD + IGF-1R-Ab group than in the HF group.
Huang [33]	2012	BALB/c-nu/nu	LNCaP xenograft	HFD (59.9%) vs. HCD (9.5%) vs CD (41.2%)	Tumor volume	The tumor growth of LNCaP xenograft was significantly higher in the HFD group than in the HCD and CD groups.
Wang [34]	2012	a) nude mice, b) Prostate-specific Pten deletion mouse	a) pten-/- allograft	High ω-6 vs ω-3 diet	a) tumor volume and weight, b) body weight, invasion rate, Ki67	ω-3 PUFA resulted in slower growth of castration-resistant tumors compared to ω-6 PUFA.
Vandelsluis [35]	2013	Nu/nu athymic mice	LNCaP xenograft	HFD (23.8%) vs. SD (6.0%)	Tumor volume	The HF with exercise group showed significantly higher tumor growth rates compared to all other groups. The SD with exercise group had significantly lower tumor growth rates of compared to the HFD without exercise group.
Pommier [36]	2013	C57BL/6 Lxra and Lxrb double knockout mice		Normal or hypercholesterolemic diet	Presence of PIN, number of Ki-67 positive cells	High-cholesterol diet induced proliferation in LXR mutant mouse prostate.
Huang [37]	2014	BALB/c-nu/nu	LNCaP xenograft	HFD (59.9%) vs. LFD (9.5%)	Tumor volume	The tumor growth of LNCaP xenograft was significantly higher in the HFD group than the LFD groups.
Moiola [38]	2014	Swiss nu/nu	PC-3 xenograft	HFD (homemade) vs. CD	Tumor volume	No significant differences in tumor growth were observed in CD-fed mice; however, we found that only 60% of HFD-fed mice inoculated with CtBP1-depleted cells developed a tumor.
Chang [39]	2014	TRAMP mouse		HFD (45%) vs. CD (10%)	Histophathologica score	Histopathological scores in the dorsal and lateral lobes were higher in the 10-week HFD group than in the 10-week CD group.
Liu [40]	2015	Pten haploinsufficientmale mice		High calorie vs. regular diet	mPIN score	High-calorie diet caused neoplastic progression, angiogenesis, inflammation, and epithelial-mesenchymal transition
Cho [41]	2015	a) TRAMP, b) C57BL/6J	b) TRAMPC2 allograft	HFD (60%) vs. CD (10%)	Rate of poorly differentiated ca, tumor weight	In TRAMP mice, HFD feeding increased the incidence of poorly differentiated carcinoma. In the allograft model, HFD increased solid tumor growth and the expression of proteins related to proliferation/angiogenesis.
Xu [42]	2015	TRAMP		HFD (40%) vs. ND (16%)	Tumor formation rate, survival	The mortality of TRAMP mice from HFD group was significantly higher than that of normal diet group (23.81% and 7.14%, *p* = 0.035). The tumor incidence of HFD TRAMP mice at 20th week was significantly higher than normal diet group (78.57% and 35.71%, *p* = 0.022)
Xu [43]	2015	TRAMP		HF (40%) vs. ND (16%)	Tumor incidence, survival	TRAMP mice in HFD group had significantly higher mortality rates than those in the normal diet group (*p* = 0.032). The HFD group had a significantly higher tumor formation rate at age 20 weeks than the normal diet group (*p* = 0.045).
Lo [44]	2016	SCID	PDX kidney capsule xenograft	HF (43%) vs LF (6%)	Pathology and biomarker expression	Prostate cancer tumorigenicity is not accelerated in the setting of diet-induced obesity or in the presence of human PPAT.
Liang [45]	2016	Immunocompetent FVB mice	MycCap alloraft	High ω-6 vs. ω-3 diet	Tumor volume	Tumor volumes were significantly smaller in the ω-3 than in the ω-6 group (*p* = 0.048).
Huang [46]	2016	BALB/c-nu/nu	LNCap xenograft	HFD (59.9%) vs LF (9.5%)	Intratumoral AKT and Extracellular Signal-regulated Kinase (ERK) activation, AMPK inactivation	HFD resulted in AKT and ERK activation and AMPK inactivation.
Labbe [47]	2016	Prostate specific Pten and Ptpn1 deletion mouse		HFD vs. chow	Microinvasive rate	PCa in Pten-/-Ptpn1-/- mice was characterized by increased cell proliferation and Akt activation, interpreted to reflect a heightened sensitivity to IGF-1 stimulation upon HFD feeding
Kwon [48]	2016	14K-creER PTEN (K14-CreER;Pten^fl/fl^;mTmG (K14-Pten-mTmG) triple transgenic mice		HFD vs. RD	PIN 3/4 rate	HFD increased the number of PIN.
Zhang [49]	2016	C57BL6	RM1 mouse prostate cancer alloglaft	HFD (58%) vs. chow	Tumor growth	CXCL1 chemokine gradient was required for the obesity-dependent tumor ASC recruitment, vascularization and tumor growth promotion
Chang [50]	2017	C57BL6		HFD (45%) vs. chow	Cav-1 secretion from adipose tissue	Cav-1 secretion was evident in adipose tissues and were substantially promoted in HFD-fed mice.
Kim [51]	2017	SCID	PC-3 xenograft	10%, 45%, or 60% fat	Tumor size, tumor weight	The 45% and 60% fat diets significantly promoted the growth of xenografts comparison to the 10% fat diet
Nara [52]	2017	a) BALB/c-nu/nu	a) LNCap xenograft, b) PC-3 and DU145 cultured with mice serum	HFD (59.9%) vs. CD (9.5%)	a) Tumor volume, b) cell proliferation	The tumor growth of prostate cancer LNCaP xenograft was significantly higher in the HFD group than in the CD groups. Cells cultured with HFD mouse serum had higher proliferation.
Huang [53]	2017	BALB/c-nu/nu	Intraperitoneal injection PC-3M-luc-C6	HFD (59.9%) vs. LF (9.5%)	Luciferase activity (IVIS), number of metastasis	HFD and PrSC increased luciferase activity and number of metastasis.
Hayashi [54]	2018	Prostate-specific Pten deletion mouse		HFD (62.2%) vs. CD (12.5%)	Tumor growth	HFD accelerated tumor growth alogn with the inflammatory response.
Massillo [55]	2018	C57BL/6J	TRAMP C1 allograft	HFD (37%) vs. CD (5%)	Tumor volume	HFD significantly increased tumor growth and serum estradiol in mice.
Chen [56]	2018	Prostate specific Pten and Pml deletion mouse		HFD (60%) vs. chow (17%)	Rate of mice having metastases	A HFD-derived metastatic progression and increases lipid abundance in prostate tumors
Hu [57]	2018	TRAMP		HFD (40%) vs. CD (16%)	Proportion of poor tumor differentiation and tumor metastasis	A trend toward poorer PCa differentiation was observed in HFD-fed mice, while no statistical significance was detected.

Abbreviations: HFD: high-fat diet, LFD: low-fat diet, HC: high-calorie diet, LC: low-calorie diet, NKCD: high-fat/no-carbohydrate ketogenic diet, WD: Western-style diet, CR: calorie restriction, Ab: antibody, SD: standard diet, CD: control diet, PDX: patient-derived xenograft, NWD: new Western-style diet.

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
