# Peer review of "Research Evidence on High-Fat Diet-Induced Prostate Cancer Development and Progression"

_jcm, 2019, doi:10.3390/jcm8050597_

Reviewer 1 Report

The authors did literature review on the topic thoroughly and the order of the review is very neat. I would recommend to modify abstract to make it clearer to the readers that the article is about the systematic review on animal model, prostate cancer cell line etc.  I was misunderstood that this is a review of human study related to prostate cancer.

Author Response

The authors did literature review on the topic thoroughly and the order of the review is very neat. I would recommend to modify abstract to make it clearer to the readers that the article is about the systematic review on animal model, prostate cancer cell line etc.  I was misunderstood that this is a review of human study related to prostate cancer.

Response: We appreciate that the reviewer recommends us to modify the abstract. As the reviewer suggested, we have corrected the abstract to clarify the article focusing on prostate cancer cell lines and animal models.

Reviewer 2 Report

Please see attached document for comments. 

Author Response

Thank you for the reviewer’s detailed reviews and many constructive suggestions. We agree with all the reviewer’s helpful comments and modified the manuscript with yellow highlighted in the revised manuscript. Please see the point-by-point responses and corrections in the revised manuscript.

Reviewer 3 Report

The title of the manuscript is “Research evidence on high-fat diet-induced prostate cancer development and progression”, but the review seems a deep description of different types of models used for the study of the relation between HFD and prostate cancers. Authors should present their observation in a more attractive manner. Moreover, they could focus their attention on relevant papers that can improve the knowledge of the readers.

 Additionally, they should analyze the relation between adipose tissue, adipokines, such as adiponectin and steroid hormones. Adipose tissue may increase for example prostate cancer risk via a dual mechanism: (i) aromatization of adrenal androgens to estrogens in the adipocyte, resulting in an increases of estrogen circulating levels (that share an important role in prostate cancer onset and progression) and consequently promoting proliferation of prostate epithelial cells; (ii) deregulation of adipokine’s expression and secretion, such as adiponectin that shows an anti-proliferative effect.

The authors have useful suggestions to improve the introduction and discussion sections by reading the following mansucripts (doi: 10.18632/oncotarget.6220; doi: 10.3389/fonc.2018.00002; doi: 10.3390/ijms20040839)

Minor concerns:

Avoid the use of acronyms  such as KO in the abstract section.

Please revise the lines 43-50 of the introduction section: there are a lot of run on sentences and two times the authors stated “this study aimed-----the review aims”

The title of the second paragraph “Difference in preclinical models” doesn’t introduce correctly the topic

Author Response

The title of the manuscript is “Research evidence on high-fat diet-induced prostate cancer development and progression”, but the review seems a deep description of different types of models used for the study of the relation between HFD and prostate cancers. Authors should present their observation in a more attractive manner. Moreover, they could focus their attention on relevant papers that can improve the knowledge of the readers.

 Additionally, they should analyze the relation between adipose tissue, adipokines, such as adiponectin and steroid hormones. Adipose tissue may increase for example prostate cancer risk via a dual mechanism: (i) aromatization of adrenal androgens to estrogens in the adipocyte, resulting in an increases of estrogen circulating levels (that share an important role in prostate cancer onset and progression) and consequently promoting proliferation of prostate epithelial cells; (ii) deregulation of adipokine’s expression and secretion, such as adiponectin that shows an anti-proliferative effect.

The authors have useful suggestions to improve the introduction and discussion sections by reading the following mansucripts (doi: 10.18632/oncotarget.6220; doi: 10.3389/fonc.2018.00002; doi: 10.3390/ijms20040839)

Response: Thank you for the reviewer’s helpful comment. We totally agree with the reviewer’s suggestion that we should discuss about the relation between adipose tissue, adipokines, such as adiponectin and steroid hormones. We add the sentences about the relation between adipose tissue, adipokines, such as adiponectin and steroid hormones with the recommended citations according to the reviewer’s recommendation (P11, 498-502)

Minor concerns:

Avoid the use of acronyms such as KO in the abstract section.

Response: As the reviewer suggested, we corrected the word.

Please revise the lines 43-50 of the introduction section: there are a lot of run on sentences and two times the authors stated “this study aimed-----the review aims”

Response: We modified the sentence based on the reviewer’s concern (P2, line 52-53).

The title of the second paragraph “Difference in preclinical models” doesn’t introduce correctly the topic

Response: As the reviewer suggested, we changed the title of second paragraph to “Various preclinical models”

Round  2

Reviewer 3 Report

Authors improved the manuscript according to my suggestions.

A list of abbreviations  can be helpful to the readers.